# A Novel Bionebulizer Approach to Study the Effects of Natural Mineral Water on a 3D In Vitro Nasal Model from Allergic Rhinitis Patients

**DOI:** 10.3390/biomedicines12020408

**Published:** 2024-02-09

**Authors:** Joana Viegas, Elsa M. Cardoso, Lucile Bonneau, Ana Filipa Esteves, Catarina L. Ferreira, Gilberto Alves, António Jorge Santos-Silva, Marco Vitale, Fernando A. Arosa, Luís Taborda-Barata

**Affiliations:** 1CICS-UBI—Health Sciences Research Centre, University of Beira Interior, Avenida Infante D. Henrique, 6200-506 Covilhã, Portugal; viegasjoana@gmail.com (J.V.); cardoso.elsamaria@fcsaude.ubi.pt (E.M.C.); lulu.2000113@gmail.com (L.B.); ana.filipa.e2@gmail.com (A.F.E.); cferreira@fcsaude.ubi.pt (C.L.F.); gilberto@fcsaude.ubi.pt (G.A.); arosa@fcsaude.ubi.pt (F.A.A.); 2ESS-IPG-School of Health Sciences, Polytechnic Institute of Guarda, Rua da Cadeia, 6300-307 Guarda, Portugal; 3Faculty of Health Sciences, University of Beira Interior, Avenida Infante D. Henrique, 6200-506 Covilhã, Portugal; ssilvaaj@gmail.com; 4Unhais da Serra Thermal Spa, Avenida das Termas, 6215-574 Unhais da Serra, Portugal; 5Faculty of Medicine and Surgery, University Vita-Salute San Raffaele, 20132 Milan, Italy; vitale.marco@hsr.it; 6FoRST—Fondazione per la Ricerca Scientifica Termale, 00198 Rome, Italy; 7UBIAir—Clinical & Experimental Lung Centre, University of Beira Interior, Estrada Municipal 506, 6200-284 Covilhã, Portugal; 8CACB—Clinical Academic Centre of Beiras, Estrada Municipal 506, 6200-284 Covilhã, Portugal; 9Department of Immunoallergology, Cova da Beira University Hospital Centre, Alameda Pêro da Covilhã, 6200-251 Covilhã, Portugal

**Keywords:** air–liquid interface, chemokines, chronic rhinitis, cytokines, histology, MucilAir^TM^, nebulizations, sulfurous thermal water, TEER, Vitrocell

## Abstract

Sulfurous thermal waters (STWs) are used as a complementary treatment for allergic rhinitis. However, there is scant data on the effects of STW on nasal epithelial cells, and in vitro models are warranted. The main aim of this study was to evaluate the dose and time effects of exposure to 3D nasal inserts (MucilAir^TM^-HF allergic rhinitis model) with STW or isotonic sodium chloride solution (ISCS) aerosols. Transepithelial electrical resistance (TEER) and histology were assessed before and after nebulizations. Chemokine/cytokine levels in the basal supernatants were assessed by enzyme-linked immunosorbent assay. The results showed that more than four daily nebulizations of four or more minutes compromised the normal epithelial integrity. In contrast, 1 or 2 min of STW or ISCS nebulizations had no toxic effect up to 3 days. No statistically significant changes in release of inflammatory chemokines MCP-1/CCL2 > IL-8/CXCL8 > MIP-1α/CCL3, no meaningful release of “alarmins” (IL-1α, IL-33), nor of anti-inflammatory IL-10 cytokine were observed. We have characterized safe time and dose conditions for aerosol nebulizations using a novel in vitro 3D nasal epithelium model of allergic rhinitis patients. This may be a suitable in vitro setup to mimic in vivo treatments of chronic rhinitis with STW upon triggering an inflammatory stimulus in the future.

## 1. Introduction

Allergic rhinitis (AR) is a chronic form of nasal inflammation that is induced by exposure to aeroallergens in a person who is sensitized [1]. The pathophysiology of chronic upper airway inflammation is multifactorial, involving genetic, environmental, and immunological factors [2,3]. In allergic rhinitis, immune interactions are complex and involve interaction between different adaptive (e.g., T helper 2 cells (Th2), regulatory T cells (Treg), B regulatory cells B cells (Breg), T follicular helper cells (Thf)), innate cells (e.g., type 2 innate lymphocytes (ILC2), dendritic cells (CD)) and epithelial cells [4]. Worldwide, complementary therapies such as inhalations with mineral thermal waters are a recognized add-on to pharmacological treatments. Some protective mechanisms have been described, such as improving local mucosa blood supply, restoring the mucosa integrity, normalizing epithelial clearance, reduction of the inflammatory states and IgE levels in the blood [5,6,7]. Despite this, there is still a limited number of published reports, and limited understanding of thermal water effects at the cellular level [8]. To bridge this knowledge gap, it is crucial to have human in vitro models that would allow functionality mimicking in vivo human studies, for instance in STW-related skin studies [9]. Three-dimensional organotypic models better mimic the complex tissue architecture and cell–cell interactions found in vivo and cells exhibit improved differentiation compared to 2D cultures, leading to the development of tissue-specific structures and functions that more closely resemble in vivo conditions. Three-dimensional models, such as the MucilAir™ nasal model, provide a better differentiation, preservation of cell–cell interactions and polarity, allowing for the assessment of drug permeability and transport in a more representative tissue context. They offer valuable insights into drug absorption and efflux mechanisms and accurate predictions of drug responses, pharmacokinetics, and toxicities, thereby reducing the reliance on animal models and improving the translatability of preclinical data. They are also used to create disease-specific tissue constructs, allowing researchers to study the effects of treatments on diseased tissues such as allergic rhinitis, and to develop personalized medicine approaches [10,11]. In addition, air–liquid interface in vitro inhalation models are increasingly being used in respiratory system research because they are more realistic as they mimic more closely the airways than classic (i.e., submerged) in vitro methods [12].

Nasal respiratory mucosa is composed of a pseudostratified columnar epithelium containing ciliary epithelial cells, goblet mucus-producing cells, and basal stem cells [13,14,15], a membrane basement, and the underlying lamina propria, comprising stromal cells such as fibroblasts and immune cells [16]. The MucilAir™ nasal model is a 3D in vitro model of the human nasal epithelium that closely mimics the biology of the nasal barrier. The model is composed of fully differentiated ciliated, goblet, and basal cells, which are arranged in a tight, polarized, pseudo-stratified epithelium. The proportion of each cell type in the model is in accordance with in vivo observations, indicating up to 50–80% of ciliated cells, 15% of goblet cells, and 5–10% of basal cells in the nasal epithelium. The model has been validated through a variety of techniques, including immunofluorescence, protein mass spectrometry, and examination of histological sections. The expression of tight junction proteins, such as ZO-1, occludin, and claudin-1, as well as the adherens junction protein E-cadherin, has been demonstrated in the MucilAir™ nasal tissues, indicating the presence of a polarized and hermetically sealed barrier, which can be evaluated by measuring transepithelial electrical resistance (TEER) [11,17].

Epithelial cells and fibroblasts are important players in AR and other allergic diseases as they are equipped to respond to allergens or pathogen-associated molecular patterns (PAMP) and take part in immune response, namely by secreting chemokines and cytokines [18,19]. These chemokines and inflammatory factors bind to the corresponding receptors and chemotactic various inflammatory cells to reach the epithelium and tissues [20]. Interleukin-8/C-X-C motif chemokine ligand 8 (IL-8/CXCL8), macrophage inflammatory protein-1 alpha/C-C motif chemokine ligand 3 (MIP-1α/CCL3), and monocyte chemoattractant protein-1/C-C motif chemokine ligand 2 (MCP-1/CCL2) are inflammatory chemokines whose main function is to recruit neutrophils, macrophages, and monocytes to the site of injury or infection, respectively [20]. In allergic rhinitis, the chemokine IL-8 has been shown to increase in nasal secretions, after allergen challenge, although it may act in connection with other chemotactic factors in the recruitment of granulocytes [21,22]. It has been reported that patients with allergic rhinitis had increased MIP-1α/CCL3 and other chemokines in nasal fluids, relative to healthy subjects, after nasal delivery of R848, a PAMP viral RNA analogue, highlighting that dysregulated innate immune responses of nasal mucosa in allergic individuals may be important in determining the outcome of viral exposures [23]. MCP-1/CCL2 was found to be highly expressed in most of the allergic rhinitis patients’ nasal samples compared to control samples of nonallergic nasal mucosa, in a microarray analysis, which can act as a recruiter of regulatory and effector CD4^+^ T and CD8^+^ T leukocytes, stimulating histamine or leukotriene release from mast cells or basophils and inducing fibrosis due to TGF-β and procollagen [24]. Interleukin-1alpha (IL-1α) is a pivotal inflammatory cytokine, released upon cell death, that acts also as an “alarmin” [25]. In patients with allergic rhinitis, this cytokine was increased in nasal lavages after nasal allergen challenge, suggesting a role in the induction and perennation of inflammatory reaction in this disease [22]. IL-33, which exhibits structural similarities with IL-1, is released by damaged or necrotic barrier cells (epithelial and endothelial cells), acting also as an “alarmin” [26], and might have a central role in the pathogenesis of allergic inflammation [27]. There is evidence that IL-10 plays a role in the pathophysiology of allergic disease [28]. IL-10 acts primarily as an anti-inflammatory cytokine, and although it is mainly secreted by regulatory T cells, some non-immune cell types, including intestinal epithelial cells, have been shown to produce this cytokine, which is critical to maintaining healthy epithelial homeostasis [29,30,31]. Murine models of ovalbumin-induced allergic rhinitis clearly showed that the cytokines/chemokines we studied are indeed expressed in vivo [32,33,34]. Overall, these mediators have been implicated in the pathophysiology of allergic rhinitis in both human and animal models of allergic rhinitis.

The main objective of this study was to evaluate the most optimal conditions for the development of a robust in vitro model mimicking exposure to respiratory crenotherapy treatments using STW. Thus, we explored the dose- and time-dependent effects of exposing 3D nasal inserts, specifically the MucilAir^TM^-HF allergic rhinitis model, to sulfurous thermal water (STW) aerosols in comparison to isotonic sodium chloride solution (ISCS) both at the epithelial integrity and at the cellular secretion of chemokines and cytokines into the basal supernatants. Our results indicate that extensive daily nebulizations compromised the integrity of nasal epithelium, irrespective of whether they were exposed to STW or ISCS. However, shorter and less frequent nebulizations appear to be non-toxic to the tissue, although with no significant impact on the release of chemokines or cytokines.

## 2. Materials and Methods

### 2.1. Sulfurous Thermal Water and Isotonic Sodium Chloride Solution

Sulfurous thermal water was collected from the thermal spa of Unhais da Serra, Portugal, every two days. The physicochemical analysis of water was performed by Laboratório de Análises do Instituto Superior Técnico, accredited by the Portuguese Institute of Accreditation (IPAC) (Lisbon, Portugal, test report No. 27677-14). The composition of the STW is reported in Table 1. Isotonic sodium chloride solution (0.9% NaCl) was acquired from Labesfal, Santiago de Besteiros, Portugal (lot number 18P2927) and was used as a control.

### 2.2. 3D Organotypic In Vitro Human Nasal Epithelial Model

Fully differentiated human nasal airway epithelium, consisting of primary epithelial cells (MucilAir^TM^, Epithelix Sàrl, Geneva, Switzerland) cocultured with human airway fibroblasts (HF) from two non-smoking patients with allergic rhinitis, was used. Fibroblasts are important for the growth and differentiation of epithelial cells (Epithelix Sàrl, Geneva, Switzerland, https://www.epithelix.com/, accessed on 9 October 2023). The information regarding each donor is shown in Table 2. The inserts were cultured on 24-well Transwells at an air–liquid interface using ready-to-use, serum-free, chemically defined MucilAir^TM^ culture medium provided by the manufacturer (containing growth factors, phenol red, and supplemented by default with antibiotics penicillin/streptomycin). The inserts were maintained in a humidified incubator (37 °C, 5% CO_2_), and the MucilAir^TM^ culture medium was changed every day during the experimental procedure. Informed consent, ethical approval, as well as quality control, were obtained and performed by the manufacturer.

### 2.3. Aerosol Generation and Exposure

Aerosol was generated from the collision between the incoming solution (Figure 1a; 1) and incoming air (Figure 1a; 2) that was pumped (Figure 1a; 3) using a BioAerosol Nebulizing Generator (Vitrocell^®^ Systems GmbH, Waldkirch, Germany). As a result, the solution was broken into small liquid droplets (approximately 0.7 to 2.5 µm), originating the aerosol. These small droplets were then distributed (Figure 1a; 4) and released into the exposure module system (Figure 1a; 5), where the MucilAir^TM^–HF nasal inserts were previously placed. The flow rate to the wells was determined by the vacuum rate that was at 2.0 ± 0.1 mL/min (Appendix A), and the temperature was kept at 37 °C through a water bath system (Figure 1a; 6). More detailed information is available on the manufacturer’s website (https://www.vitrocell.com/, accessed on 23 October 2023).

### 2.4. Experimental Design

The MucilAir^TM^–HF nasal inserts were placed in culture and exposed to STW or ISCS aerosols. The nebulizations were performed for 1, 2, 4, 8, or 15 min and the maximum number of nebulizations was 10 (Table 3 and Figure 1b). In some experiments, additional control tests were performed: exposure to clean air (CA for 2 min) and no exposure (incubator negative control). For chemokines/cytokines determination, cell supernatants were collected 24 h after each previous nebulization, and stored at −20 °C.

### 2.5. Tissue Integrity Monitoring—Transepithelial Electrical Resistance

Airway epithelium integrity was measured by transepithelial electrical resistance (TEER) before (day 0) and 24 h after each exposure. First, 200 µL of MucilAir^TM^ culture medium was added to the apical surface of each insert, and then the measurements were carried out with an EVOM2 voltohmmeter with an STX2/chopstick electrode (World Precision Instruments, Sarasota, FL, USA), in agreement with manufacturer’s recommendations. The resistance value appeared on the EVOMX screen and TEER (Ω·cm^2^) was then calculated using the following Formulas (1) and (2) [35].
R_insert_ = R_total_ − R_blank_
(1)
TEER (Ω·cm^2^) = (R_insert_ − R_membrane_) × A (2)
where R is resistance, Ω; R_membrane_ is membrane resistance = 100 Ω; and A is the surface area of the porous membrane of the insert = 0.33 cm^2^. According to Epithelix Sàrl, TEER values were typically in the range of 200–600 Ω·cm^2^. Afterward, the culture medium was gently aspirated from the apical surface and MucilAir^TM^–HF inserts were immediately exposed to STW, ISCS, or clean air. Inserts that were not exposed were returned to the incubator.

### 2.6. Morphology Monitoring—Histological Evaluation 

The inserts were fixed in 3.7–4.0% formaldehyde solution for 30 min at room temperature. After the dehydration process, the inserts were embedded in paraffin and sectioned at 4 µm with a microtome (Microm HM 340E, Microm, Walldorf, Germany). Hematoxylin and Eosin (H&E) staining was used to highlight the nasal epithelial histological integrity. Slides were viewed using an Olympus BX41 phase contrast and darkfield microscope (Olympus, Tokyo, Japan), an Axiocam 105 color camera (Zeiss, Oberkochen, Germany), and ZEN 2.3 lite Microscopy Software (Zeiss).

### 2.7. Determination of Chemokine/Cytokine Levels

Levels of chemokines (IL-8/CXCL8, MCP1/CCL2 and MIP-1α/CCL3) and cytokines (IL-1α, IL-33, and IL-10) in the basal supernatants of MucilAir^TM^–HF nasal inserts were assessed by sandwich enzyme-linked immunosorbent assay (ELISA) kits from Boster (Boster Biotech, Pleasanton, CA, USA). The measurements were performed according to the manufacturer’s instructions and specific sensitivity levels can be found in the manufacturer’s manuals. The average zero standard O.D. was subtracted for each sample and standard. Standard curves were generated with a four-parametric logistic (4-PL) curve fit and data were analyzed using MyAssays Analysis Software Solutions (www.myassays.com, accessed on 9 June 2023). Detection limits were automatically calculated by the MyAssays Software, and the mean percentage of recoveries from the calibrator standards were 95.65%, 108.09%, 100.56%, 101.90%, 106.60%, 98.9% for IL-8/CXCL8, MCP1/CCL2 and MIP-1α/CCL3, IL-1α, IL-33, and IL-10, respectively. All 4-PL curves had a R^2^ > 0.9.

### 2.8. Statistical Analysis

Statistical analysis was conducted using Prism 9.0 software (GraphPad Software, San Diego, CA, USA) or SPSS 28 software (IBM, Armonk, NY, USA), and *p* < 0.05 was considered to indicate a statistically significant difference. Data were checked for normality using the Kolmogorov–Smirnov test. TEER values and normalized production of IL-8/CXCL8 were expressed as the mean ± standard error of the mean (SEM) and were analyzed using parametric tests: one-way ANOVA with multiple comparisons Dunnet’s test (versus day 0) or one-way ANOVA with multiple comparisons Sídák test (STW versus ISCS, for each time point). Since the cytokine levels were not normally distributed, these data are expressed as the median (minimum and maximum, interquartile range/IQR, range) and were analyzed using the nonparametric Kruskal–Wallis and Dunn’s multiple comparisons test (only for IL-8/CXCL8 and MIP-1α/CCL3; for the remaining chemokines/cytokines only descriptive data are shown since many samples were outside (below or above) the calibrator standard curve range).

## 3. Results

### 3.1. Tissue Integrity Monitoring

Nasal inserts from AR patients were subjected to daily nebulizations of STW or ISCS (Figure 1), for time-course and dose-response experiments. Tissue integrity was monitored by measuring TEER (Figure 2) and by analyzing tissue histology (Figure 3).

Our first approach considered that typical thermal water treatment of allergic rhinitis patients in Portuguese thermal spas includes 15 min of daily nebulizations for 14 days. Accordingly, in a pilot study, inserts were exposed daily for 15 min for 10 days (Table 3), with STW (*n* = 6) or ISCS (*n* = 5), using a low flow rate of 2 mL/min, recommended by the Vitrocell manufacturer. However, tissue integrity monitored by measuring TEER revealed that this condition was too aggressive (TEER < 100 Ω·cm^2^) for the 24-well inserts already at day 4. Thus, in the next set of experiments, we decreased the period of nebulizations (ranging from 8 to 1 min), as well as the number of nebulizations (ranging from seven to three), as shown in Table 3. As can be seen in Figure 2, aerosol exposure for 8 min was also destructive after five (STW) or six (ISCS) nebulizations (TEER < 200 Ω·cm^2^). In these conditions, TEER values were significantly lower than basal TEER values at day 0, i.e., no nebulizations (*p* < 0.05, one-way ANOVA with multiple comparisons Dunnet’s test). Comparing nebulizations with STW versus ISCS showed lower TEER values after five nebulizations for 8 min with STW (*p* < 0.0001, one-way ANOVA with multiple comparisons Sídák test). Thus, in the second set of experiments, this condition (8 min) was abandoned. Exposures for 4 min also revealed a statistically significant decrease in the TEER values for the ISCS, after six nebulizations (*p* < 0.05, one-way ANOVA with multiple comparisons Dunnet’s test). Although nebulizations for 4 min with STW maintained the epithelial TEER values rather constant, histological evaluation on day 5 showed that most epithelial tissues lost the typical pseudostratified morphology (Figure 3a). The same kind of metaplasia could also be observed on day 5 in inserts exposed for 1 or 2 min, both with STW and with ISCS (Figure 3a,b, respectively). Clean air or no nebulization (incubator control) maintained tissue integrity (Figure 3c). Overall, evaluating both TEER and histology, we conclude that up to three nebulizations for 1 or 2 min preserves tissue integrity, using STW or ISCS.

### 3.2. Chemokines/Cytokines’ Release into the Basal Supernatant

Next, we wanted to evaluate whether nebulization with STW or with ISCS control influenced chemokine/cytokine profile secretion into the basal medium. We evaluated the levels of three pro-inflammatory chemokines, IL-8/CXCL8, MCP-1/CCL2, and MIP-1α/CCL3, two “alarmins” that are released upon cell death, IL-33 and IL-1α, and one anti-inflammatory cytokine, IL-10.

We detected the production of IL-8/CXCL8 in all samples of allergic rhinitis inserts (*n* = 112) with median (minimum–maximum; IQR; range) values of 708.2 (139.7–1173.0; 723.3; 1033.3) pg/mL, without statistically significant differences among the different tested conditions (*p* > 0.05, Kruskal–Wallis and Dunn’s multiple comparisons test). To correct for an observed batch effect between set 3 and set 4, we normalized the production of IL-8/CXCL8 relative to day 0. Figure 4 shows that the levels of IL-8/CXCL8 released into the basal supernatants after 24 h remained constant, independently of the number of nebulizations, or the type/duration of each nebulization (STW versus ISCS/1 versus 2 min, one-way ANOVA with multiple comparisons Sídák test, *p* > 0.05).

Regarding MCP-1/CCL2, all samples analyzed were also positive. However, it is important to call attention to two limitations. First, although we could retrieve concentration values, most of the values were outside and above the range of the standards (15.6 to 1000 pg/mL). Second, 30 out of 108 analyzed samples were even above the curve fit (ceiling effect), and no concentration could be estimated. Therefore, the values presented here are an underestimation of the real production of MCP-1/CCL2 for 24 h, with a median (minimum–maximum; IQR; range) value of 599.8 (125.9–7292.0; 1053.9; 7166.1) pg/mL (*n* = 78), and no statistical analysis was applied.

In all supernatants of the assessed inserts (*n* = 36, set 3), we detected low levels of MIP-1α/CCL3: median (minimum–maximum; IQR; range) value of 77.9 (63.0–149.6; 14.3; 86.6) pg/mL, but no statistically significant differences were observed among the groups (*p* > 0.05, Kruskal–Wallis and Dunn’s multiple comparisons test).

For IL-33, IL-1α, and IL-10, most of the samples were below the lower limit of quantification (floor effect). Therefore, no statistical analysis was performed. IL-33 was evaluated in 112 inserts and only 3 inserts were positive (65.2, 260.2, and 301.5 pg/mL). IL-1α was below detectable levels in all assessed inserts (*n* = 40, set 3). Regarding IL-10 (*n* = 40, set 3), only 9 inserts were positive, median (minimum–maximum; IQR; range) value of 2.6 (0.4–7.4; 4.9; 7.0) pg/mL, while the remaining 31 inserts were below detection level.

## 4. Discussion

To the best of our knowledge, this is the first investigation studying the effects of using a prolonged and repeated exposure regime of STW or ISCS on nasal MucilAir^TM^-HF inserts.

One of the central findings of this study is the relationship between the dose and duration of aerosol exposure and the integrity of nasal epithelial cells. The results indicate that more than four daily nebulizations lasting four minutes or more compromised the normal epithelial integrity, as assessed by TEER and histological analysis. This observation suggests a potential threshold beyond which extended exposure to aerosols, whether STW or ISCS, can lead to detrimental effects on the nasal epithelial barrier. Comparison with in vivo settings is difficult for several reasons. The simulation of the flow inside a nasal cavity is complex due to its anatomic geometry [36,37,38], and modeling inevitably requires simplifications and assumptions [39]. Another factor that influences the total flow rate (minute respiratory volume) is the respiratory rate, which is about 12 breaths per minute. Adopting these assumptions, flow rates range from 6 L/min [40], 12 L/min [41], 17 L/min to 34 L/min [38], to 60 L/min or 70 L/min in non-invasive high-flow-rate therapy with humidified medical gas [42] have been described. To estimate the flow rate in an in vivo 0.33 cm^2^ nasal area, it is also necessary to consider additional assumptions, namely that the flow rate is the same through all nasal areas. Estimations of the total surface area of both nasal cavities are about 160 cm^2^ [15] or 150 cm^2^ [43], and if the paranasal sinuses are included, about 400 cm^2^ [41]. Considering all these assumptions, we conclude that in our study of the 0.33 cm^2^ insert, the applied flow rate was 6.0× to 80.0× lower than in vivo settings (Appendix A), using the low flow rate of 2 mL/min recommended by the manufacturer. Accordingly, it was notable that shorter exposures, specifically 1 or 2 min of nebulization with either STW or ISCS, had no toxic effects on the tissue for up to three days. This implies that there exists a range of safe conditions in terms of exposure duration, within which the epithelial integrity remains intact. These findings offer valuable guidance for future applications of aerosol nebulizations, helping to establish limits to ensure the preservation of nasal tissue health in vitro setups.

Numerous studies have investigated the simulation of nebulized thermal water exposure in the nasal cavity [44], and in vivo experiments in humans have consistently demonstrated the anti-inflammatory effects of thermal water nebulization in various respiratory diseases [8,45,46,47,48]. In situations whereas an inflammatory state is present, sulfur compounds such as hydrogen sulfide may shape adaptive immune Th and B-cell response [49]. For instance, reduction of IgE has been previous reported in in vivo human studies [7,46]. Whether STW is accomplished by a polarization into a protective shift towards IgGs remains to be investigated [50]. In vitro studies with thermal water consist mostly of applications in skin models [51,52,53,54]. However, to date, there has been a notable absence of research on the in vitro nebulization of thermal water using a three-dimensional (3D) model of the nasal epithelium. To our knowledge, no prior investigations have explored the nebulization of thermal water in in vitro nasal epithelium. Moreover, existing in vitro studies involving thermal water have primarily exposed monolayer cells, such as A549 cells [55], to thermal water within the culture medium. Notably, these studies have not utilized a 3D nasal epithelium model for examining the effects of thermal water nebulization. Conversely, the 3D nasal epithelium model has been employed in numerous studies to investigate the exposure and nebulization of various compounds, including nanoparticles [56,57,58,59,60,61], pollens [62], pollutants [57,63,64], pharmaceutical compounds [65,66], and viral infections [67,68]. However, in these studies, the compounds were either added to the culture medium or applied to the apical side in the form of a solid aerosol, or liquid solution [69,70].

We did not detect a significant presence of two “alarmins”, IL-1α and IL-33, known to be released upon epithelial cell death [25,26]. In an experimental allergic rhinitis mouse model, it was shown that IL-33 protein was constitutively expressed in the nucleus of nasal epithelial cells and was promptly released in response to nasal exposure to ragweed pollen [71]. IL-1α production was shown to be associated with antigen-induced late nasal response in patients with allergic rhinitis [22]. Therefore, it is possible that although we have studied nasal tissue reconstituted from cells removed from an allergic rhinitis patient, the tissue was not in an inflamed phase. A retainment of Th2-type memory has been previously described in polyp basal progenitor cells from patients with chronic rhinosinusitis, but not in non-polyp steam cells [14]. In agreement with the hypothesis that our epithelial tissue was in a non-exacerbated phase is the fact that the levels of the studied inflammatory chemokines were like, or below, the levels detected in the basal medium of human nasal or bronchial MucilAir derived from normal healthy donors [58,62,72,73,74,75]. Inflammatory stimuli such as LPS or TNF-α induce much higher levels of these chemokines [62,75]. This is a limitation of this study, and future studies should include positive inflammatory controls like LPS, TNF-α, IL-1β or house dust mite allergen extract to demonstrate the model’s sensitivity to the known stimuli. Some minor differences in the concentration of chemokines among studies may depend on the technique used or even inter-individual variations. We did not find IL-10 production in our system, although this anti-inflammatory cytokine has been implicated in maintaining healthy epithelial homeostasis [29,30,31]. In future studies aiming to investigate the potential anti-inflammatory effect of STW [6] on in vitro nasal epithelial models from allergic rhinitis patients, it is important that there is a prior challenge to cells either with inflammatory or with allergens stimuli. Another limitation of our in vitro model is the absence of immune cells. Consequently, the immunological responses portrayed by this model may not be entirely representative of the more complex inter-cellular crosstalk’s. Future opportunities to compensate for the complex interaction among epithelial and immune cells might be addressed in vitro using co-cultures, i.e., incorporating macrophages or lymphocytes [62,76]. This might provide a starting point to gather insights on the mechanisms how STW or STW-derived formulations would affect the epithelial barrier in allergic rhinitis before validation in human trials.

A final limitation was that although we used a high number of nasal inserts, they were all developed from a single donor. This has also been the situation with many other published studies using nasal inserts, given the complexity of their preparation, and which have used only 1–3 donors [61,72,77,78].

## 5. Conclusions

To conclude, the characterization of the chemokine/cytokine profile and the identification of safe time and dose conditions in these in vitro aerosol nebulizations (not more than three once daily 1- or 2 min-long exposures, at 2 mL/min), provide a promising avenue for future research and the development of therapeutic strategies in the context of chronic allergic rhinitis. Nonetheless, further studies with more donors should be performed to confirm this pilot study.

## Figures and Tables

**Figure 1 biomedicines-12-00408-f001:**
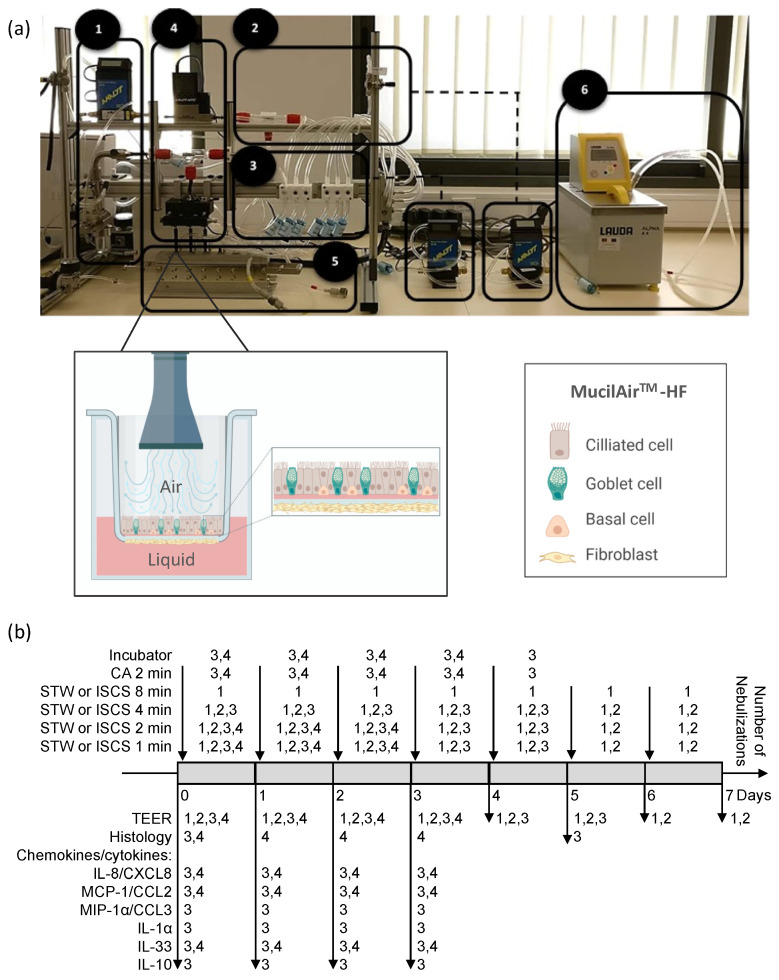
Experimental setup of the air-liquid interface 3D in vitro model. (**a**). Vitrocell^®^ exposure system and air–liquid interface overview: 1—generator system; 2—clean air distribution system; 3—vacuum system; 4—distribution system; 5—exposure module system and schematic view of the air–liquid interface, where reconstructed human 3D nasal pseudostratified epithelium, comprising four cell types (ciliated, goblet, basal and fibroblasts—MucilAir^TM^-HF) were exposed to aerosol; and 6—water bath system. (**b**). Experimental design of the four sets of experiments (1, 2, 3, and 4; see also Table 3). CA, clean air control; IL-1α, interleukin-1alpha; IL-8/CXCL8, interleukin-8/C-X-C motif chemokine ligand 8; IL-10, interleukin-10; IL-33, interleukin-33; ISCS, isotonic sodium chloride solution; MCP-1/CCL2, monocyte chemoattractant protein-1/C-C motif chemokine ligand 2; MIP-1α/CCL3, macrophage inflammatory protein-1 alpha/C-C motif chemokine ligand 3; STW, sulfurous thermal water; TEER, transepithelial electrical resistance. Created with BioRender.com.

**Figure 2 biomedicines-12-00408-f002:**
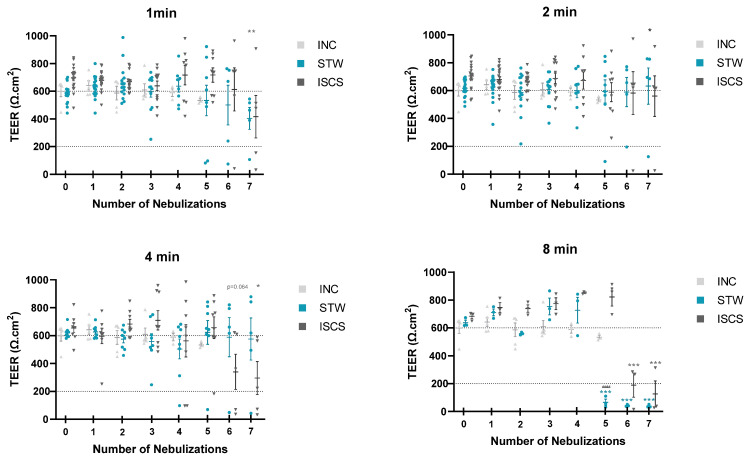
Assessment of tissue integrity by transepithelial electric resistance. Effect of repeated nebulizations (up to 7 days) on the TEER values of MucilAir^TM^-HF inserts (Total *n* = 98, see Table 3) with STW (during 1, 2, 4, or 8 min), or ISCS (also during 1, 2, 4 or 8 min). Data are presented as mean ± standard error of the mean (SEM). Horizontal dash lines indicate the typical healthy TEER values for MucilAir^TM^-HF inserts, in the range 200–600 Ω·cm^2^. * *p* ≤ 0.05, ** *p* ≤ 0.005, *** *p* ≤ 0.001, one-way ANOVA with multiple comparisons Dunnet’s test versus day 0; #### *p* ≤ 0.0001, one-way ANOVA with multiple comparisons Sídák test versus ISCS). INC, incubator control; ISCS, isotonic sodium chloride solution; min, minutes; STW, sulfurous thermal water; TEER, transepithelial electrical resistance.

**Figure 3 biomedicines-12-00408-f003:**
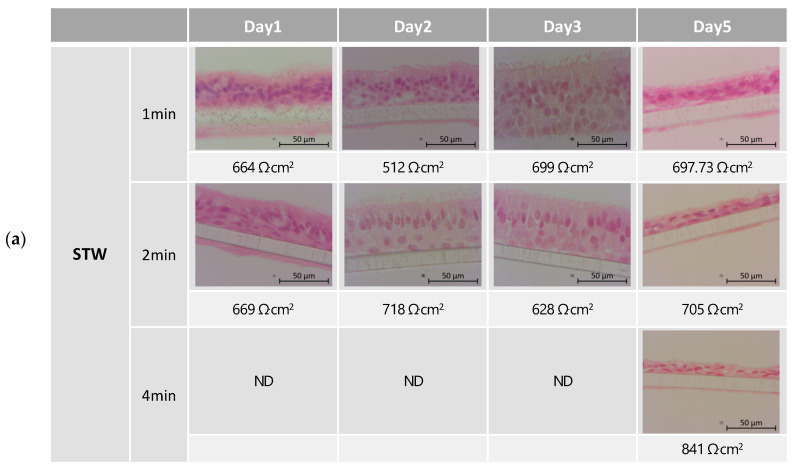
Morphology monitoring by histological evaluation. Representative histological images of human MucilAir^TM^-HF nasal inserts harvested at the end of day 0, day 1, day 2, day 3, or day 5. (**a**) Nebulizations with STW during 1, 2, or 4 min. (**b**) Nebulizations with ISCS for 1, 2, or 4 min. (**c**) Incubator (no nebulization) or exposure to clean air for 2 min. For each insert the respective TEER value is shown. Fixed tissue slices stained with hematoxylin and eosin. In some inserts, human fibroblasts or the Transwell membrane detached during the tissue processing. CA, Clean Air; Inc, incubator; ISCS, isotonic sodium chloride solution; min, minutes; ND, not done; STW, sulfurous thermal water; TEER, transepithelial electrical resistance.

**Figure 4 biomedicines-12-00408-f004:**
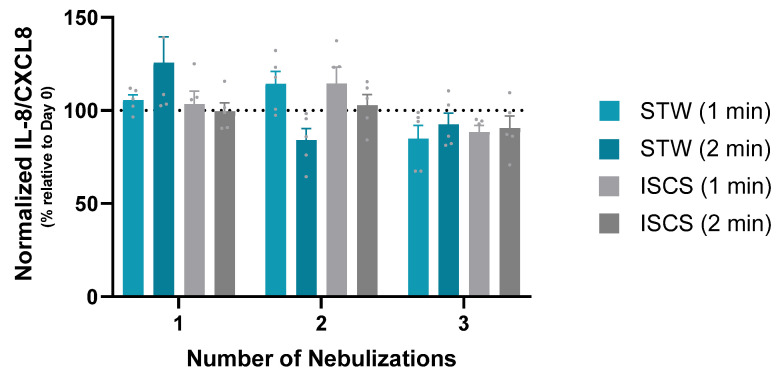
Normalized production of IL-8/CXCL8. The levels of IL-8/CXCL8 present in the basal supernatants 24 h after 1, 2, or 3 nebulizations of MucilAir^TM^-HF inserts (*n* = 5, 2 from set 3, and 3 from set 4) with STW (during 1 or 2 min), or ISCS (also during 1 or 2 min), were normalized to day 0 (no exposition) for each insert. The dashed line represents day 0 (100%). Data are presented as mean ± standard error of the mean (SEM). One-way ANOVA with multiple comparisons Sídák test. IL-8/CXCL8, interleukin-8/C-X-C motif chemokine ligand 8; ISCS, isotonic sodium chloride solution; min, minutes; STW, sulfurous thermal water.

**Table 1 biomedicines-12-00408-t001:** Physicochemical composition of sulfurous thermal water used in this study.

Parameter	Result		Method
Physico-chemical			
Temperature at source	37.5 °C		
pH (at 24 °C)	8.43		SMEWW 4500 H+
Conductivity (at 20 °C)	303 µS·cm^−1^		NP EN 27888:1996
Resistivity	3.3 × 10^3^ Ω·cm		LAE 4.3 A
Total sulfur	0.14 mmol·L^−1^		M.M. (CI)
Total sulfuring of sulfide	13 mL (I2 0.01 N·L^−1^)		M.M. 3.11 (21 May 2013)
Hydrogen sulfide	<0.5 mg (H_2_S/L)		M.M. 2.2.7 (7 February 2003)
Total alkalinization	75.5 mg (CaCO_3_)·L^−1^		SMEWW 2320
Hardness	10 mg (CaCO_3_)·L^−1^		SMEWW 2340B
Silica (SO_2_)	55 mg (SiO_2_)·L^−1^		SMEWW 4500 Si-C
Total silicon	57 mg (SiO_2_)·L^−1^		SMEWW 4500 Si-C
Dry residue	226 mg·L^−1^		SMEWW 1030 E
Total mineralization	268 mg·L^−1^		M.M. 2.1.11 (3 April 2009)
Anions			
Bicarbonate (HCO_3_^−^)	82.9 mg (HCO_3_)·L^−1^	1.36 mEq·L^−1^	M.M. 2.2.7 (7 February 2003)
Carbonate	<2 mg (CO_3_)·L^−1^	___	M.M. 2.2.7 (7 February 2003)
Chloride (Cl^−^)	26 mg·L^−1^	0.73 mEq·L^−1^	SMEWW 4110B
Fluoride (F^−^)	16 mg·L^−1^	0.84 mEq·L^−1^	SMEWW 4110B
Hydrosulfide	2.2 mg (HS)·L^−1^	0.07 mEq·L^−1^	M.M. 2.2.7
Silicate	3.4 mg (H_3_SiO_4_)·L^−1^		M.M. 2.2.7 (7 February 2003)
Nitrate	<0.3 mg (NO_3_)·L^−1^	___	SMEWW 4110B
Nitrite	<0.010 mg (NO_2_)·L^−1^	___	SMEWW 4500 NO2-B
Silicate	3.4 mg (H_3_SiO_4_)·L^−1^	0.04 mEq·L^−1^	M.M. 2.2.7 (7 February 2003)
Sulphate (SO_4_^2−^)	7.9 mg (SO_4_^2−^)·L^−1^	0.16 mEq·L^−1^	SMEWW 4110 B
Cations			
Ammonia nitrogen	0.08 mg (NH_4_)·L^−1^	___	M.M. 4.1 (22 November 1997)
Calcium (Ca^2+^)	3.9 mg·L^−1^	0.19 mEq·L^−1^	EPA 300.7:1986
Lithium	0.3 mg·L^−1^	0.04 mEq·L^−1^	EPA 300.7:1986
Magnesium	0.15 mg·L^−1^	0.01 mEq·L^−1^	EPA 300.7:1986
Sodium (Na^+^)	67 mg·L^−1^	2.91 mEq·L^−1^	EPA 300.7:1986
Potassium (K^+^)	2.0 mg·L^−1^	0.05 mEq·L^−1^	EPA 300.7:1986
Iron	<0.006 mg·L^−1^	___	M.M. 5.4 (EAA-CG) (6 May 2013)

**Table 2 biomedicines-12-00408-t002:** Donors’ information of the nasal MucilAir^TM^-HF allergic rhinitis model used in this study.

Identification Code	Batch Number	Age (Years Old)	Sex	Smoker	Origin	Pathology	Viral Status *
EP29	HF-MD006201	36	Male	No	Caucasian	Allergic rhinitis	Negative
EP14	HF-MD041901	52	Female	No	Caucasian	Allergic rhinitis	Negative

* HIV-1 (anti-HIV-1 antibodies), HIV-2 (anti-HIV-2 antibodies), hepatitis B (HBs antigen and anti-HBc antibody), hepatitis C (anti-HCV antibody), mycoplasma (mycoplasma detection with MycoAlert^®^, Lonza, Walkersville, MD, USA).

**Table 3 biomedicines-12-00408-t003:** General information of the five sets of experimental designs.

Set	ID Code	Number of Nebulizations	Duration of Nebulization (Minutes)	Condition (*n*)			
				STW	ISCS	CA	None (Incubator)
Pilot	EP29	10	15	6	5	-	NA
1	EP14	7	1	3	3	-	NA
			2	3	3	-	NA
			4	3	3	-	NA
			8	3	3	-	NA
2	EP14	7	1	2	2	-	NA
			2	2	2	-	NA
			4	2	2	-	NA
3	EP14	0	NA	NA	NA	NA	3
		5	1	3	3	-	NA
			2	3	3	2	NA
			4	3	3	-	NA
4	EP14	0	NA	NA	NA	NA	2
		3	1	11	11	-	NA
			2	11	11	3	NA

CA, clean air; ID, identification; ISCS, isotonic sodium chloride solution; *n*, number of inserts; STW, sulfurous thermal water; NA, not applicable; -, not performed.

## Data Availability

The data presented in this study are available on request from the corresponding author.

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
