# Peer review of "A Novel Bionebulizer Approach to Study the Effects of Natural Mineral Water on a 3D In Vitro Nasal Model from Allergic Rhinitis Patients"

_biomedicines, 2024, doi:10.3390/biomedicines12020408_

Round 1

Reviewer 1 Report

Comments and Suggestions for Authors

Authors examined the effects of sulphurus thermal water and an isotonic NaCl solution on an in vitro nasal model of allergic rhinitis. Although they described their methodology in details they didn’t provide references and examples of the use of this model (MucilAir-HF) for nasal pathologies.

If their purpose was to study the suitability of this model in nasal pathology then more comparisons with in vivo models are required e.g. do the concentrations of the cytokines studied follow the same pattern etc. If their purpose was to compare sulphurous thermal water with NaCl 0.9% then I am not sure if their statistically insignificant results warrant publication.

In general their idea and their model is interesting, however the purpose of the research is not clear and their conclusions are not optimally supported by their results

Author Response

Reviewer 1 (Round 1)

  1. “Authors examined the effects of sulphurus thermal water and an isotonic NaCl solution on an in vitro nasal model of allergic rhinitis. Although they described their methodology in details they didn’t provide references and examples of the use of this model (MucilAir-HF) for nasal pathologies.”

Reply: More information (references and examples) of the use of this model (MucilAir-HF) for nasal pathologies was added in the introduction (lines 52-66) and in the discussion (lines 419-426). However, in these studies, the compounds were either added to the culture medium or applied to the apical side in the form of a solid particles, or direct liquid solution, not as liquid aerosol droplets.

  1. “If their purpose was to study the suitability of this model in nasal pathology then more comparisons with in vivo models are required e.g. do the concentrations of the cytokines studied follow the same pattern etc. If their purpose was to compare sulphurous thermal water with NaCl 0.9% then I am not sure if their statistically insignificant results warrant publication.”

Reply: We thank the reviewer to prompt us to clarify this issue:

  1. a) The main objective of the current study was to set up the most optimal conditions for testing the biological effects of bionebulization with sulphurous NMW on the nasal mucosa of patients with allergic rhinitis. Apart from testing epithelia integrity (using histology and TEER) we also wanted to see whether there was any preliminary effect of cytokine/chemokine production in this model. If we compare the cytokines we studied, we will be able to highlight that there are murine models of Ovalbumin-induced allergic rhinitis, which clearly showed that the cytokines/chemokines we studied are indeed expressed in vivo (lines 114-115). Thus, this suggests that our model is well aligned with what is seen in in vivo models of allergic rhinitis. We have expanded and justified further the selection of this chemokine/cytokine panel (lines 91-118) citing new references studding these factors in nasal fluids in AR.
  2. b) Our nebulization exposure model required that we would follow an approach that would be akin to the most robust clinical studies, namely randomized, placebo-controlled clinical trials. Such clinical trials which have been performed in patients with chronic rhinitis do involve comparison of respiratory crenotherapy using NMW against crenotherapy using isotonic saline (some examples are: Doi: 10.4193/Rhin09.065, and Doi: 10.1016/j.amjoto.2010.02.004).

We would also like to add that, as mentioned above (point#1), the MucilAir-HF model has been used mostly to evaluate solid aerosols, but not liquid-derived aerosols. To the best of our knowledge, this is the first study demonstrating time- and dose-evaluation of exposure to liquid aerosol droplets. Accordingly, although we have tested both STW and ISCS, the primary objective of this study was to establish time- and dose- safe conditions for nebulizations of the 3D epithelium.

Finally, we believe that, although preliminary, our study brings novel and significant results for the scientific community that can serve as a starting point and guidance for future in vitro and in vivo studies. The pilot character of this study is emphasized in the conclusion (lines 470-474).

  1. “In general their idea and their model is interesting, however the purpose of the research is not clear and their conclusions are not optimally supported by their results”

Reply: We thank the reviewer for this comment, which has helped us to better clarify the objective of the study. In particular we believe we have now made it clearer that the main objective was to evaluate the most optimal conditions for the development of a robust in vitro model mimicking exposure to respiratory crenotherapy treatments using STW. Thus, we used dose- and time-experiments for the exposure of 3D nasal inserts (MucilAirTM-HF-Allergic Rhinitis) to STW or isotonic sodium chloride solution (ISCS) control aerosols. We have better clarified this information in the abstract (line 20), as well as at the end of the Introduction (lines 119-121) to improve match/support the results obtained.

Reviewer 2 Report

Comments and Suggestions for Authors

Introduction:
- Provide more detailed background on the pathophysiology of allergic rhinitis and the role of epithelial cells and innate immune responses. Explain how thermal waters are thought to modulate inflammation. cite PMID:27852425.

- Discuss limitations of conventional 2D culture models for studying epithelial responses. Highlight advantages of 3D organotypic models like better differentiation, preservation of cell-cell interactions and polarity.
- Discuss possible nasal anomalies influencing mucociliar clearance as septal deviation or concha bullosa. cite doi:10.1016/j.amjoto.2021.103368.

- Specifically describe the MucilAir model composition, structure and validation data to justify its relevance for mimicking the nasal epithelium.

Methods:
- Report demographic details like age, sex, atopy history, symptom severity, medication use, season of sampling. This provides context for generalizability.
STW composition:
- Give a table listing all constituents analyzed (e.g. sulfur compounds, minerals, cation concentrations etc.) along with quantitative values. Reference the analytical methods. 
- Include a negative control like cell culture media alone to establish baseline responses.
- Use positive inflammatory controls like IL-1β, LPS or house dust mite allergen extract to demonstrate the model's sensitivity to known stimuli. Report validation data.
Cytokine assays:
- Provide details on assay formats (sandwich ELISA?), detection limits, percent recoveries from quality controls.  
- Justify selection of analytes based on literature evidence for their role in allergic rhinitis.
Statistics:  
- Describe specific statistical tests used for each experiment/endpoint.
- For non-parametric data, report median and IQR/range along with statistically significant p-values.

Results:
- For TEER and histology, provide representative micrographs to support quantitative data.
- Report TEER and cytokine levels at each timepoint rather than just changes, for transparency.  
- Perform statistics to compare STW vs ISCS effects at individual timepoints rather than just over time.
- Normalize cytokine levels to relevant controls (e.g. media alone) and positive controls where used.  
- Mention any outliers, distribution of data (spread) and ceiling/floor effects encountered.

Discussion:  
- Critically discuss causes for epithelial disruption at higher doses, citing literature on toxic thresholds.

- discuss the genetic role of mucociliary disorders, ionic channels, epigenetic and risk of rhinosinusitis. cite doi:10.1111/coa.13870

- Compare responses to other thermal waters/aerosols studied in previous models.
- Acknowledge limitations in extrapolating extremely high aerosol flows to human physiology.  cite PMID:23821760.
- Propose mechanisms for STW effects based on constituents, citing relevant studies. cite doi:10.1016/j.jaci.2017.09.052.
- Suggest most clinically relevant safe conditions for future in vitro or clinical studies.
- Recommend optimal duration/spacing of treatments based on recovery time of epithelial barrier.
- Highlight knowledge gaps and future scope - e.g. studying inflammatory challenge, immunological responses.
- Temper conclusions since only single donor data and acknowledge need for replication.
- Discuss translational potential and path forward - e.g. validation in human trials, new formulations.

Comments on the Quality of English Language

no

Author Response

REPLIES TO REVIEWERS’ COMMENTS

Reviewer 2 (Round 1)

“Introduction:”

  1. - Provide more detailed background on the pathophysiology of allergic rhinitis and the role of epithelial cells and innate immune responses. Explain how thermal waters are thought to modulate inflammation. cite PMID:27852425.”

Reply: We have provided a more detailed background on the pathophysiology of allergic rhinitis and the role of epithelial cells and innate immune responses (lines 38-43), as well as how thermal waters are thought to have a protective role, including in modulating inflammation (lines 45-48). New citations were added, including PMID:27852425.

  1. “- Discuss limitations of conventional 2D culture models for studying epithelial responses. Highlight advantages of 3D organotypic models like better differentiation, preservation of cell-cell interactions and polarity.”

Reply: As suggested, we have added this information (lines 52-66).

  1. “- Discuss possible nasal anomalies influencing mucociliar clearance as septal deviation or concha bullosa. cite doi:10.1016/j.amjoto.2021.103368.”

Reply: We thank the reviewer for this suggestion but we believe that although the influence of nasal anomalies influencing mucociliar clearance is interesting in clinical situations, this aspect falls out of the scope of this manuscript. In fact, in this model various aspects that may have to do with mucociliary clearance may be studied (namely ciliary beating time), but this aspect requires additional equipment for its analysis. Since we wanted to concentrate on studying epithelial integrity (histology, TEER) and cytokine/chemokine production, we did not add any aspect regarding the study of epithelial ciliary beating time. However, in the future, we believe that this aspect may also be analysed using this model. 

  1. “- Specifically describe the MucilAir model composition, structure and validation data to justify its relevance for mimicking the nasal epithelium.”

Reply: As requested, we have added this information (lines 70-81), and thank the reviewer for this suggestion.

“Methods:”

  1. – “Report demographic details like age, sex, atopy history, symptom severity, medication use, season of sampling. This provides context for generalizability.”

Reply: Although we agree that such information (atopy history, symptom severity, medication use, season of sampling) would be of added value, the fact that we have used commercial inserts, does not allow us to have access beyond the information that appears in the certificate analysis provided by Epithelix. Therefore, we were only able to add some more additional data to Table 2 (Pathology, and viral status), but believe that these data are provide information that is aligned with what was requested by the reviewer.

STW composition:

  1. “- Give a table listing all constituents analyzed (e.g. sulfur compounds, minerals, cation concentrations etc.) along with quantitative values. Reference the analytical methods.”

Reply: As requested by the reviewer, we have expanded Table 1 to list all constituents analyzed (separated by anions and cations), quantitative values, and analytical methods, accessible in the bulletin analysis provided by the analysis laboratory of the Instituto Superior Técnico (accredited by the Portuguese Institute of Accreditation (IPAC).

  1. “- Include a negative control like cell culture media alone to establish baseline responses.”

Reply: We would like to thank the reviewer for allowing us to clarify this issue. Our negative control like cell culture media alone was the “Incubator” control. We have clarified this in M&M (line 182)

  1. “- Use positive inflammatory controls like IL-1β, LPS or house dust mite allergen extract to demonstrate the model's sensitivity to known stimuli. Report validation data.”

Reply: We really thank the reviewer for this very important point. The main objective of the study was to assess the capacity of the in vitro model to allow studying nasal epithelial integrity and cytokine production and analyse the effects of STW on such parameters in a baseline model. Thus, as a baseline model, this objective was attained. We now plan to start analyzing the effect on a further stimulated epithelium, using Cytomix. In fact, we already have some preliminary results that show that the nasal epithelial inserts can reduce TEER to lower levels, and enhance cytokine production (IL-8, and MCP-1) when exposed to Cytomix. However, these results still need to be further confirmed and will be produced in the context of a future approach focusing on nasal epithelial responses to various types of pro-inflammatory stimuli. 

Thus, although the suggestion of the referee is very interesting, unfortunately we cannot currently adequately provide the requested information. We have stressed further this point that in the discussion. (lines 422-444), and hope to be able to subsequently address this issue in our future studies, once we have completed the stimulation approaches we have described.

“Cytokine assays:”

  1. – Provide details on assay formats (sandwich ELISA?), detection limits, percent recoveries from quality controls.”

Reply: As requested by the reviewer, details on the assay formats, detection limits and percentage of recoveries from quality controls were introduced in the M&M section (lines 220, and 226-229). Regarding the quality controls, the kits used do not provide a positive control, but we have added the percentage of recovery from the Calibrator Standards. We have also added de R2 values for the goodness of fit measurement. All curves had a R2>0.9 (lines 223-226).

  1. “- Justify selection of analytes based on literature evidence for their role in allergic rhinitis.”

Reply: The chemokines (IL-8/CXCL8, MCP-1/CCL2, MIP-1α/CCL3) and cytokines (IL-1α, IL-33, IL-10) that we studied, were selected as a panel that included both inflammatory, anti-inflammatory and alarmins families, and because they have been studied and implicated in the pathophysiology of allergic rhinitis, based on evidence from various studies evaluating their roles in immune cell recruitment, inflammation, and immune regulation within the nasal mucosa of affected individuals. These molecules are therefore considered relevant targets for assessing production and for understanding the underlying mechanisms of allergic rhinitis. New references were integrated in the penultimate paragraph of the introduction (lines 91-118).

“Statistics:” 

  1. - Describe specific statistical tests used for each experiment/endpoint.”

Reply: This information was available at M&M section and at the legend of each figure. However, following the reviewer’s suggestion, and to make it clearer, we have also added this information throughout the text, next to each experiment/endpoint.

  1. “- For non-parametric data, report median and IQR/range along with statistically significant p-values.”

Reply: As requested by the reviewer, we have added IQR/range for all data involving cytokines. Non-parametric statistical analysis was only performed for IL-8/CXCL8 and MIP-1α/CCL3, and no statistically significant differences were observed. For the other cytokines, due to the reported ceiling/floor effects we indicate only the descriptive statistics. Unfortunately, we had no more media to test other dilutions (ceiling). This was now highlighted in the reviewed manuscript.

“Results:”

  1. - For TEER and histology, provide representative micrographs to support quantitative data.”

Reply: We believe that the answer to this point can be found in Figure 3 of the manuscript. However, in case we did not adequately understand the concern, we ask the reviewer for a clarification.

  1. - Report TEER and cytokine levels at each timepoint rather than just changes, for transparency.

Reply: In order to comply with this relevant request by the reviewer, we have prepared a new Figure 2, with TEER values at each time points, instead of mean values. Although we feel this new version is more difficult to follow due to the high number of points, it does increase its transparency. Regarding IL-8 levels, we reported changes rather than individual levels because there was an obvious batch effect between set 3 and set 4, that would jeopardize a global evaluation.

  1. “- Perform statistics to compare STW vs ISCS effects at individual timepoints rather than just over time.”

Reply: We would like to clarify that TEER comparison between STW vs ISCS was done at individual time points using the Sídák test, and the only statistically significant differences are marked in Figure 2. We have added this information in the main text (lines 318-319).

  1. “- Normalize cytokine levels to relevant controls (e.g. media alone) and positive controls where used.”

Reply: By normalizing IL-8 to day 0, it was a normalization against media alone. The quantification of IL-8 in the incubator control was only determined at day 0 and day 3, therefore it was not appropriate to use as control for normalization (due to the inexistence of IL-8 determination in incubator control at day 1 and 2). No statistically significant differences among the different tested conditions, including the incubator controls were observed (p>0.05, Kruskal-Wallis and Dunn's multiple comparisons test). As mentioned above (see point #8), we have not done positive controls.

  1. “- Mention any outliers, distribution of data (spread) and ceiling/floor effects encountered.”

Reply: We thank the reviewer for this important request. These aspects have now been taken into consideration, and we have clarified them further in the results section (lines 365, 373, 374).

“Discussion:” 

  1. - Critically discuss causes for epithelial disruption at higher doses, citing literature on toxic thresholds.”

Reply: We have now better clarified this aspect, since we detected an error in the flow rate that had been described, as mentioned at the beginning of the cover letter. In fact, the actual flow rate that was used to expose the nasal tissue inserts was 2 mL/min (which was the recommended value). From our experiments, we believe that we have shown that epithelial disruption may occur when a higher number of repetitive nebulizations were performed or with higher nebulization exposure times, rather than due to the flow that was used, since this was kept stable. These aspects were highlighted at the top of the cover letter and discussion was changed accordingly.

  1. “- discuss the genetic role of mucociliary disorders, ionic channels, epigenetic and risk of rhinosinusitis. cite doi:10.1111/coa.13870”

Reply: We thank the reviewer for this suggestion but found that these aspects, although quite relevant would divert the attention from the main focus of our paper. In any case, we have added some information on the role of genetic factors on development of chronic upper airway inflammation in the introduction instead (lines 38-39)

  1. “- Compare responses to other thermal waters/aerosols studied in previous models.”

Reply: We agree that this aspect is quite important, we have significantly enhanced information about this topic in the discussion (lines 418-431).

  1. “- Acknowledge limitations in extrapolating extremely high aerosol flows to human physiology. cite PMID:23821760.”

Reply: We believe that this situation is no longer applicable, since we have corrected this aspect in the paper, with the actual flow reaching the epithelial cells being 2mL/min, and not 2L/min, as we had miswritten. Please see point #18.

  1. “- Propose mechanisms for STW effects based on constituents, citing relevant studies. cite doi:10.1016/j.jaci.2017.09.052.”

Reply: We agree that this is a fundamental aspect and have added information on some mechanisms for STW-associated effects based on constituents; the suggested paper was cited (lines 411-418).

  1. “- Suggest most clinically relevant safe conditions for future in vitro or clinical studies.”

Reply: We agree with the reviewer that these aspects are quite relevant, and have added suggestions for safe in vitro liquid STW aerosol exposure conditions (lines 471-472); however, we believe these aspects cannot be directly transposable to clinical studies, and, therefore, we opted not to comment regarding such clinical aspects.

  1. “- Recommend optimal duration/spacing of treatments based on recovery time of epithelial barrier.”

Reply: We also agree that this aspect is relevant, and have added information regarding the optimal duration (lines 471-472); however, we did not study spacing of treatments, since we followed the clinical routine of daily treatments that are used at thermal spas.

  1. “- Highlight knowledge gaps and future scope - e.g. studying inflammatory challenge, immunological responses.”

Reply: In the discussion we have emphasized the limitations of the present study, and future opportunities to overcome knowledge gaps (lines 444-446; 453-460)

  1. “- Temper conclusions since only single donor data and acknowledge need for replication.”

Reply: As requested, we have now tempered the conclusions, highlighting the need for replication. (lines 474-475), and have also added a paragraph at the end of the discussion (lines 464-467).

  1. “- Discuss translational potential and path forward - e.g. validation in human trials, new formulations.”

Reply: We thank the reviewer for this important suggestion, and have inserted some pertinent aspects regarding this issue into the discussion (lines 460-463).

Round 2

Reviewer 1 Report

Comments and Suggestions for Authors

I believe that the authors have addressed my concerns and their purpose and methodology are now more clearly presented